# Defining and Assessing Quality in IoT Environments: A Survey

**Aggeliki Sgora [1,*] and Periklis Chatzimisios [2,3]**

1  Department of Digital Media and Communication, Ionian University, 28100 Argostoli, Greece
2  Department of Information and Electronic Systems Engineering, International Hellenic University of Greece (IHU), Alexander Campus, 574 00 Sindos, Greece
3  Department of Electrical and Computer Engineering, The University of New Mexico, Albuquerque, NM 87131-0001, USA
*  Correspondence: asgora@ionio.gr; Tel.: +30-26710-27131

**Abstract:** With the proliferation of multimedia services, Quality of Experience (QoE) has gained a lot of attention. QoE ties together the users' needs and expectations to multimedia application and network performance. However, in various Internet of Things (IoT) applications such as healthcare, surveillance systems, traffic monitoring, etc., human feedback can be limited or infeasible. Moreover, for immersive augmented and virtual reality, as well as other mulsemedia applications, the evaluation in terms of quality cannot only focus on the sight and hearing senses. Therefore, the traditional QoE definition and approaches for evaluating multimedia services might not be suitable for the IoT paradigm, and more quality metrics are required in order to evaluate the quality in IoT. In this paper, we review existing quality definitions, quality influence factors (IFs) and assessment approaches for IoT. This paper also introduces challenges in the area of quality assessment for the IoT paradigm.

**Keywords:** QoS; QoE; IoT; QoD; QoI; mulsemedia; fog computing; QoE influence factors





## 1. Introduction

Quality of Service (QoS), according to the International Telecommunications Union (ITU), is "the totality of characteristics of a telecommunications service that bear on its ability to satisfy stated and implied needs of the user of the service" [1]. It comprises both network-related performance (e.g., bit error rate, latency) and non-network related performance (e.g., service provisioning time, different tariffs) [2]. Thus, in order to satisfy the users' needs, for more than 30 years the telecommunications industry, as well as academia, investigated several mechanisms in order to guarantee QoS in the provided telecommunication services.

However, with the exponential growth of the video-based services, the telecom operators realized that catering the quality expectation of the end users in multimedia services is the most important parameter [3]. Humans are considered quality meters, and their expectations, perceptions, and needs with respect to a particular product, service, or application carry great value [4,5].

The user experience of multimedia applications is inevitably bounded up with the notion of Quality of Experience (QoE) [6]. Lagjhari et al. described QoE as "the blueprint of all human quality needs and expectations" [4].

However, with the introduction of the Internet of Things (IoT), traditional terms and approaches used for defined or evaluating services may not be suitable/sufficient for the IoT context [7], where consumers may no longer be users but machines. Moreover, the QoE requirements in such a heterogeneous environment can vary with respect to the considered IoT application domain; even QoE requirements among IoT applications of the same IoT domain may vary [8]. Furthermore, due to the fact that, in IoT, decisions are taken based on data infusion from multiple sensors in case of failures, the effects are often multidimensional [9].

Currently, there is no standardization or set of best practices as to how the subjective tests can be conducted and even if there was, it would be practically infeasible to carry out subjective tests for every existing as well as new applications [7]. Moreover, existing subjective methodologies do not consider QoE influence factors (IFs) of the IoT environment, such as the usefulness of the application [7].

Thus, in such heterogeneous environments, existing quality related issues that were initially tailored for humans, such as QoE definitions, evaluation approaches and provision mechanisms, should be re-examined in order to check their validity in a machine-to machine environment. In addition, in cases where human interaction is not required, the traditional definition of QoE is not valid, and new metrics to evaluate the quality in IoT environments are required.

In the literature, several survey papers may be found that focus on IoT, e.g., [10,11]) or QoE, e.g., [12–14]). However, there are few survey papers that examine QoE in IoT environments.

Fizza et al. [15] reviewed existing QoE definitions and QoE models for autonomic IoT. However, the authors suggested only one definition for QoE in IoT. In addition, regarding the QoE modeling, the authors provided limited information concerning the role of data in the QoE evaluation of the IoT applications. Moreover, the authors in [7] focused on the QoE IFs and presented a QoE taxononomy for IoT, while Bures et al. [16] consolidated the IoT quality characteristics into a unified view. A survey concerning QoE evaluation for autonomic IoT applications can be found in [17].

The contributions of the current paper are as follows:

- Review existing definitions of QoE that are suitable for IoT environments, since nowadays new terms have been introduced to define and evaluate the quality of IoT applications.
- Identify and categorize the quality IFs for IoT. More specifically, we have collected and classified IFs that may found into the literature and are necessary for the creation of a successful quality model for IoT.
- Review existing quality assessment approaches for IoT applications.

The rest of this paper is organized as follows: Section 1 presents an overview of IoT and fog computing architectures. Section 2 reviews existing quality definitions that are suitable for IoT, while Section 3 overviews and categorizes quality indicators for IoT. Section 4 reviews existing QoE models and frameworks for IoT. Section 5 discusses challenges in the area of QoE assessment in the IoT context, while Section 6 concludes the paper.

## 2. Internet of Things (IoT) and Fog Computing

The term "Internet of Things" (IoT) was coined by Kevin Ashton in 1999 to describe the ability of network objects (connected to the Internet) to bring new services [18]. Since then, the IoT paradigm has gained a great momentum. More specifically, according to the statistics portal Statista (www.statista.com (accessed on 1 June 2022)), the number of IoT connected devices is expected to rise more than 75 billion in 2025 [19]. Figure 1 illustrates several fields of IoT applications including transportation, healthcare, home automation and smart cities [20]. However, these diverse IoT applications and devices from various manufacturers create such network heterogeneity that a unified and inter-operable standard is very difficult to achieve [10].

Currently, there is no global consensus on the architecture of IoT, thus, many different IoT architectures may be found in the literature [11]. The basic architecture has three layers [10]:

- A things layer (also known as perception, device or sensor layer) that consists of the sensing hardware, and its main objective is to interconnect things in the IoT network.
- A middle layer (also known as transport layer) that processes the received data from the things layer and determines the optimum data transmission path to the IoT servers.

•   An application layer (also known as the business layer) that provides information management, data mining, data analytics, decision-making services, as well as the required services to end-user or machines.

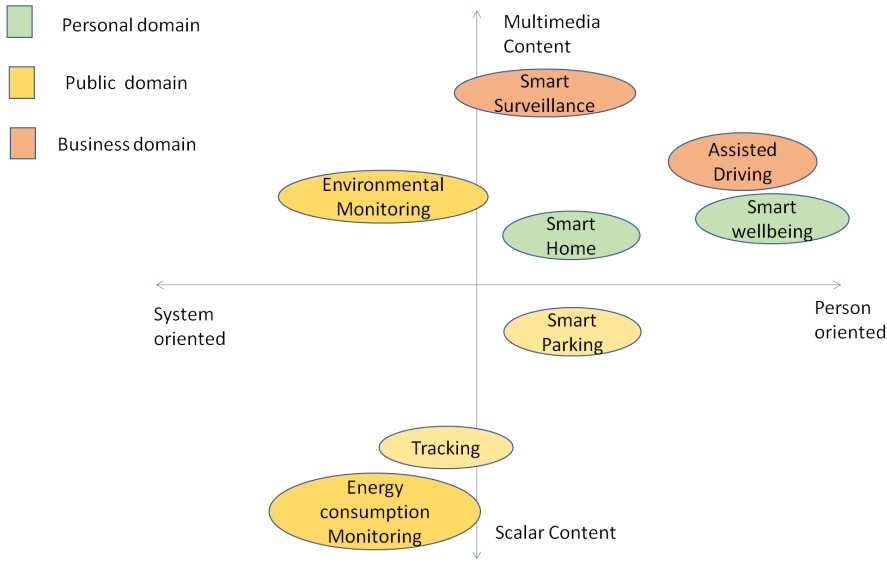

**Figure 1.** An IoT applications' classification.

However, according to Kassab et al. [11] the superior model with respect to the elements is the architecture proposed by Yousefpour et al. (2018) [21], a fog computing architecture. Fog computing is a computing paradigm (introduced by Cisco) that deals with the requirements of time-sensitive IoT applications [22]. The idea is that instead of processing the sensor data on the cloud, to address this issue at the edge [23]. By doing so, the following advantages are accomplished [22]: i) applications are executed closer to end-user and IoT devices, (ii) performance metrics for real-time applications such as latency, response time, and cloud workload are enhanced, (iii) network scalability is increased, and (iv) device mobility is supported.

Figure 2 depicts the three layers of the fog computing architecture. The lowest layer consists of the IoT devices that produce massive data and potentially are heterogeneous, geographically distributed and have mobility features [24]. The fog computing layer is composed by the fog nodes, intelligent intermediate devices from different networking components [25] and retransmits the workload to the cloud servers at given time intervals.

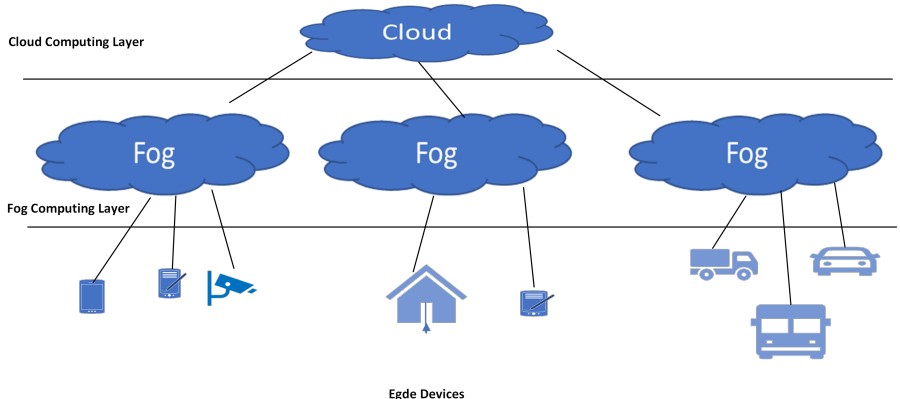

**Figure 2.** The fog computing architecture.

### 3. Quality in a IoT Environment

In telecommunications, the most suitable metric to assess end-to-end quality is QoE. The most frequently used definition for QoE is the one given by the ITU [26], where it is defined as "The overall acceptability of an application or service, as perceived subjectively by the end-user". Moreover, since many researchers pointed out that the inclusion of the term "acceptability" as the basis for a QoE definition is not ideal, during the Dagstuhl Seminar in 2009, the term acceptability was newly defined as "the outcome of a decision [yes/no] which is partially based on the Quality of Experience" [27]. However, even with this modification, the definition still follows a user-centric approach; thus, it does not reflect the machines' perspective.

Another popular definition of QoE is the one described in the Qualinet White paper [28] in which QoE is defined as "the degree of delight or annoyance of the user of an application or service. It results from the fulfillment of his or her expectations with respect to the utility and/or enjoyment of the application or service in the light of the user's personality and current state". Raake and Eggger [27] extended the definition of the Qualinet White paper in order to also include the term system. Thus, according to the new definition QoE is "the degree of delight or annoyance of a person whose experiencing involves an application, service, or system. It results from the person's evaluation of the fulfillment of his or her expectations and needs with respect to the utility and/or enjoyment in the light of the person's context, personality and current state".

The Qualinet's definition according to Floris and Atzori [20] is valid for general multimedia applications/services and, thus, it can be used for cases where humans are the recipients of the content provided by multimedia IoT applications.

However, in the IoT context, where exist applications that do not require any human intervention, such as smart parking, connected vehicles, etc., the term QoE cannot be used to describe quality. To this end, several researchers introduced new terms to define quality in the IoT domain. Mivoski et al. [29] introduced the term Quality of IoT-experience (QoIoT), which aggregates the delivered quality of an IoT service from the perspective of both humans and machines within the context of autonomous vehicles. More specifically, the QoIoT metric compromises the traditional user-centric QoE metric and the Quality of Machine Experience (QoME), an objective metric that "measures the quality and performance of intelligent machines and their decisions".

Karaadi et al. [30] defined the term "Quality of Things" (QoT) for multimedia communications in IoT to express the quality of fulfilling an IoT task in a Multimedia IoT (M-IoT) [31]. However, the authors do not provide any measurement methodology.

Rahman et al. [9] defined the term Quality of Systems (QoSys), an objective metric like QoE, that measures "the quality and performance of the Systems of Systems (SoS), and the decisions made by those". Thus, the metric $QoE_{IOT}$ is introduced in order to evaluate the quality in an IoT scenario from the perspective of both humans and machines.

Wang et al. [32] introduced the term quality of X (QoX), as a comprehensive evaluation metric that combines QoS, QoE, Quality of Data (QoD) and Quality of Information (QoI).

In this end, Fizza et al. [17] introduce the term Quality of autonomic IoT applications as "an aggregate quantitative value of various IoT quality metrics measured at each stage of the autonomic IoT application life cycle".

Table 1 overviews and highlights the specific drawbacks of each definition. As can be seen, there is no definition that can generally express the end-to-end quality in IoT environments.

**Table 1.** Quality and IoT.

| Paper | Recipient User | Recipient Machine | Term | Shortcoming |
|---|---|---|---|---|
| | | | | Too generic definition. |
| [9] | | x | QoT | It is not clear how it can be measured |
| [17] | x | x | QoE$_{AIoT}$ | Autonomic IoT systems |
| [26] | x | | QoE | It does not reflect the machine's focused quality |
| [27] | x | | QoE | It does not reflect the machine's focused quality |
| [28] | x | | QoE | It does not reflect the machine's focused quality |
| [29] | x | x | QoIoT | It cannot be applied to Autonomic IoT systems |
| [30] | x | x | QoE$_{IoT}$ | It cannot be applied to Autonomic IoT systems |

## 4. Key Quality Indicators for IoTs

As stated in [23], the first step in creating a successful quality model is to create a taxonomy of its influence factors (IFs). However, identifying these factors is not an easy task to accomplish.

As concerns QoE, existing approaches classify IFs into multiple dimensions. Stankiewicz et al. [3 identified that the factors that impact the QoE of multimedia services are QoS, Grade of Service (GoS), Quality of Resilience (QoR), as well as several orthogonal factors depicted in Figure 3.

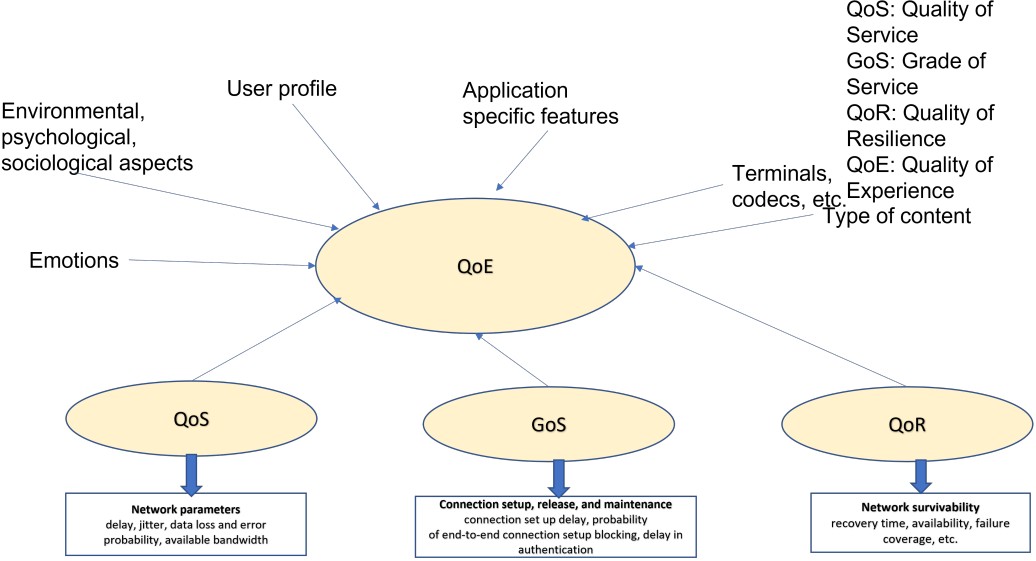

**Figure 3.** Factors influencing QoE [33].

Le Callet et al. [28] classify the QoE IFs into the following 3 categories [5]:

1.  Human IFs that present any variant or invariant property or characteristic of a human user (e.g., motivation, gender, age, education, etc.);
2.  System IFs that refer to properties and characteristics that determine the technically produced quality of an application or service (e.g., QoS, display size, resolution);
3.  Context IFs that embrace any situational property to describe the user's environment in terms of physical, temporal, social, economic, task, and technical characteristics (e.g., day of time, cost, etc.).

However, since in IoT the data acquired by devices (objects), as well as the information acquired and processed are important parameters, two more categories/dimensions may be found in the literature: the Quality of Data (QoD) that is used for data quality evaluation,

and the Quality of Information (QoI) that is used for information quality evaluation. However, in several papers, the term QoI is used to determine the quality of information or data [34,35].

Table 2 overviews the most common QoD metrics, while Table 3 shows the most common QoI metrics.

**Table 2.** QoD metrics.

| Metric | Definition |
| --- | --- |
| Completeness | The extent to which data are of sufficient breadth, depth and scope for the task at hand [18] |
| Precision | The extent to which the collected data are precise |
| Truthfulness | The extent to which the collected data are from reliable resource [19] |
| Accuracy | The extent to which data are correct and accepted |
| Usefulness | The extent to which the sensed data are for the application [15] |
| Consistency | The extent to which data are presented in the same format and compatible with previous data [18] |
| Timeliness | The extent to which data are valid for decision making [15] |

**Table 3.** QoI metrics.

| Metric | Definition |
| --- | --- |
| Recall | Proportion of relevant information retrieved from a query [9] |
| Detail | Completeness of the information provided to the decision-maker [9] |
| Validity | Provided information is true or not [19] |
| Accuracy | Accuracy degree of information to the decision maker [19] |
| Timeliness | Timely information for an IoT service (opposite to latency) [19] |
| Precision | How close the measured values are to each other [19] |

Fizza et al. [17] considered the following IFs for the autonomic IoT applications: (1) the QoD; (2) the Quality of Device (QoDe); (3) the QoS; (4) the Quality of Context (QoC); (5) the QoI (6) Quality of Security and Privacy (QoSe & P); and (7) Quality of Actuation (QoA).

Rahman et al. [9] also considered the Quality of Cost (QoC) due to the fact that the machines use some resources in terms of computation, storage, or energy, and such consumptions should be optimized.

Ikeda et al. [36] considered two sets of metrics: physical metrics emerging in the IoT architecture, such as network QoS, sensing quality, and computation quality, and metaphysical metrics demanded by users, such as accuracy, context and timeliness.

Pal et al. [7] classified the QoE IFs for IoT environments into three distinct categories:

1. Technical, which represent the various QoS factors, which are popular in the multimedia context and also relevant with the IoT examined scenario.
2. User, which represent the subjective characteristics of the users of the IoT applications.
3. Context, which are related to the data and information quality along with specific application requirements that can vary depending upon the usage scenario.

Figure 4 presents the taxonomy for IoT environments by Pal et al. [7].

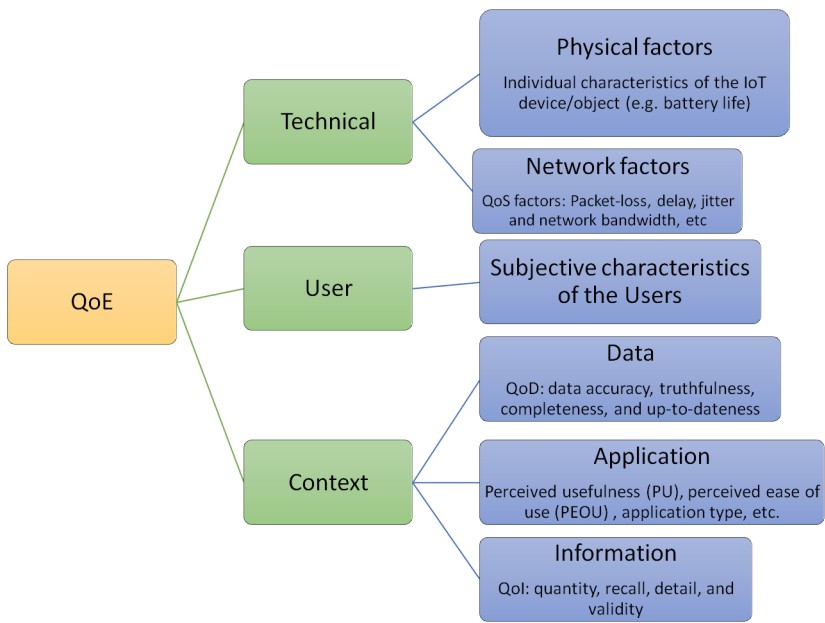

**Figure 4.** QoE taxonomy for IoT [7].

Nashaat et al. [23] consider 3 dimensions: the Environment runtime context, the Application, and the User expectations. These factors, in addition to QoS feedback, influence the total QoE of the user by a valuable weight, as Figure 5 depicts.



**Figure 5.** QoE IFs in the IoT context [23].

Besides the QoE taxonomies for IoT applications, researchers have proposed various QoE taxonomies for specific IoT verticals. For example, Damaj et al. [37] in their taxonomy for the context of Connected and Autonomous Electric Vehicles (CAEVs) have identified several performance indicators that were grouped into categories. These categories were then mapped to 4 QoE IFs. Table 4 presents the categories and the corresponding QoE IF.

**Table 4.** QoE IFs for CAEVs [6].

| QoE IF | Categories |
| --- | --- |
| Context | Travel Efficiency, Operability |
| Cost | Affordability |
| QoS | Energy, Security, Networking and Connectivity, Survivability, Subsystem Performance |
| Human | Safety, Personal Usability |

Finally, Wette et al. [38] listed the most common IoT QoE IFs that are used in a generic Machine QoE (M-QoE) framework developed by Ericsson to accurate predict the QoE of the IoT subscribers. The selected IFs constitute different QoS, QoD and QoC metrics.

An overview of the different quality metrics are presented in Table 5.

**Table 5.** Quality IFs for IoT.

| Quality IF | Description | Refs. |
|---|---|---|
| Quality of Experience (QoE)/ Human feedback | Evaluates the overall acceptability of an application or service or system as perceived subjectively by users | [7,15,30,38] |
| Quality of Context (QoC) | Evaluates the context of the environment or the application | [7,17,23] |
| Quality of Cost (QoCo) | Evaluates the cost in terms of of computation, storage, or energy of an IoT application | [9] |
| Quality of Information (QoI) | Evaluates the quality of information | [9,15,17,19,34] |
| Quality of Data (QoD) | Evaluates the quality of data | [15,17–19] |
| Quality of Service (QoS) | Evaluates the network's capability to provide satisfied service levels | [15,17,23,33,38] |
| Quality of Device (QoDe) | Evaluates the quality of the physical IoT devices | [17] |
| Quality of Actuation (QoA) | Evaluates the correctness of the decision making/ actuation performed by an IoT application | [17] |
| Quality of Security and Privacy (QoSe & P) | Evaluates the security and privacy of an IoT application | [17] |

## 5. Quality Models for IoTs

Traditionally, qualitative methods that focus on voice perceptibility for applications usability are used for the QoE evaluation [39]. The QoE for multimedia services is evaluated by subjective, objective and hybrid assessment (a combination of both the subjective and objective approaches) [40].

In this context, a few studies that focus on modeling the relationship between human experience and quality perception in relation to the smart-wearable segment may be found in the literature [6,41,42]. QoE is considered a very important aspect of multiple sensorial media (mulsemedia) [43].

Shin et al. [41] examined the relation of users' experience and the quality perception in IoT. To achieve this goal the authors utilized a combination of qualitative and quantitative methods. Figure 6 shows the proposed QoE model in which, besides the user's behavior, coolness, satisfaction and affordance are considered as QoE factors in the IoT context.

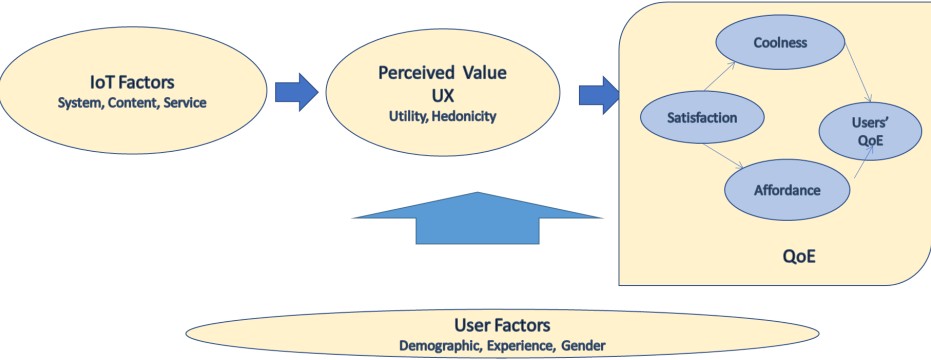

**Figure 6.** QoE IFs in the IoT context [7].

Pal et al. [44] proposed a QoE model that maps QoD and QoI to QoE. More specifically, the authors, in order to create the model, collect data from 5 wearable devices. Half of the data set is used to build the model, while the other half is used to test accuracy. The step-counts and heart-rate measurement readings by the wearables are used as QoD parameters, whereas the perceived ease of use, perceived usefulness, and richness in information are used as QoI parameters. The accuracy of their model is evaluated by comparing the QoE

obtained from the mathematical model and a subjective test with 40 participants. The authors adopted the Mean Opinion Score (MOS) to quantify the user experience.

Saleme et al. [6] studied the impact that human factors such as gender, age, prior computing experience, airflow intensity and smell sensitivity have on 360° mulsemedia QoE. A total of 48 participants (27 male, 21 female) participated in this study with ages between 16 and 65 years old. Results showed that all these factors influence the users' QoE. Guidelines for evaluating wearables' quality of experience in a mulsemedia context can be found in [43].

In addition to the QoE evaluation for wearables, several attempts have been made in order to create QoE models for IoT. One of the first attempts was made by Wu et al. [45] that calculated the overall QoE by combining two parameters: profit (expressed in terms of QoD, QoI, QoE) and cost (expressed in terms of resource efficiency, i.e., device utilization efficiency, computational efficiency, energy efficiency, storage efficiency). The same approach was also followed in several other studies illustrated in Table 6.

Another way of quantifying QoE is the layer-based approach [23], in which each layer focuses on a specific QoE IF (domain), so that the overall quality can be computed as a combination of all IFs (domains). Several layered-QoE models may be found in the related literature.

For example, Floris and Atzori [20] proposed a layered QoE model that aims to evaluate the contributions of each IF to estimate the overall QoE in Multimedia IoT (MIoT) applications [23]. More specifically, the proposed model consists of five layers: physical devices, network, combination, application, and context. The authors, in order to demonstrate the generalization of the their framework, have applied it in two use cases: a) vehicles remote monitoring and b) smart surveillance.

A similar approach is also presented in [36]. More specifically, in this framework, the physical metrics are organized into four layers (device, network, computing, and user interface) while the metaphysical metrics are organized in two layers (information and comfort). However, no evaluation of the proposed framework is provided in this work.

**Table 6.** Generic QoE calculations comparison.

| Paper | QoE | QoD | QoI | QoC | Testing |
|-------|-----|-----|-----|-----|---------|
| [9] | Evaluated from factors of 3 level-architecture (access, communication, computation, application) | Accuracy, truthfulness, completeness, up-to-dateness, precision | Timeliness, validity, recall, accuracy, detail | Computational efficiency, energy efficiency, storage efficiency | No |
| [19] | Delay, jitter, packet delivery rate, throughput, and gateway availability. | Completeness, precision, truthfulness | Quality, precision, recall, accuracy, detail, timeliness, validity | Energy consumption, interface use | Simulation |
| [45] | Evaluated from factors of 4 level-architecture (access, communication, computation) | Accuracy, truthfulness, completeness, up-to-dateness, quantity, precision | Recall, accuracy, detail, timeliness, validity | Device utilization efficiency, computational efficiency, energy efficiency, storage efficiency | No |

Suryanegara et al. [46] proposed a different approach to measure the QoE of IoT services. More specifically, their proposed framework is based on the following steps: (1) Setting up the focus of the IoT services to formulate the QoE parameters, (2) Judging the institutional users who are the users of IoT services, (3) Conducting a Mean Opinion Score (MOS) survey of IoT service users, (4) Calculating the differential MOS as the Absolute Category Rating with Hidden References (ACR-HR) quantitative scale, and (5) Providing the strategic implications to those responsible for the implementation. The authors, in order to validate the proposed framework, conducted a subjective test in Jakarta where 6 institutional users expressed their experience of utilizing IoT technology in their relevant services, i.e., managing public transportation, garbage trucks, ambulances, the fire and rescue brigade, street lighting, and water level measurement.

Finally, [17] proposed a framework to measure the quality of autonomic IoT by mapping five IoT quality metrics to the IoT application life cycle stages: (1) Data Sensing, (2) Sensed Data Transmission, (3) Data Analytics, (4) Analyzed Information Transmission, and (5) Actuation. However, approaches on how to model and measure these IoT IFs are still an open issue.

## 6. Discussion

Defining quality in an IoT environment it is not an easy task. Although several terms are proposed in the literature, with the heterogeneity of the IoT components, it is difficult to have a generic definition for quality in IoT. A specific domain definition seems to be a more appropriate solution as in [17,29]; however, a classification based on the different characteristics of the IoT is required.

Additionally, the diversity of IoT applications makes the identification of the appropriate IFs a very challenging task. In Section 3, we have collected all the Quality IFs that can be found in the literature, as Figure 7 depicts. However, the answer to the question "which IFs should be considered for this IoT application" cannot be easily provided.

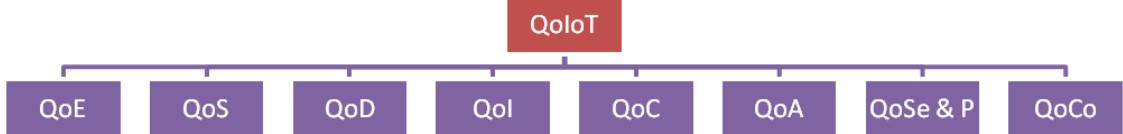

**Figure 7.** Quality IFs in the IoT context.

Machine Learning (ML) techniques can be beneficial to address this challenge, since they can be applied to predict the type of IoT application and, as a sequence, the appropriate IFs. Saovapakhiran et al. [47] in their proposed QoE-driven IoT architecture propose the use of ML techniques to tailor QoE at the User level from User Engagement metrics. However, they do not provide a quantitative solution.

In addition, the fact that existing IoT architectures are multi-tier systems increases the complexity of the measurements in IoT. Since each tier has different aspects for the quality IFs, it is not often clear where the IFs should be collected. For example, in the M-QoE framework [38], the IFs are measured for: (1) the IoT device, (2) the Radio Access Network (RAN), (3) the edge network, (4) the core network/services network and (5) the vertical slices and service layers. However, acquiring data from different tiers can result in deterioration of communication delays [47]. To deal with this issue, Saovapakhiran et al. [47] suggested the creation of different QoE domains and the local estimation of QoE in each domain. However, no implementation details are given. In addition, the fact there is a standardized architecture the QoE domains may differ according to the proposed architecture.

Furthermore, security and privacy are crucial challenges to be addressed in IoT architectures. For example, in wearable environments, as more data are collected for the QoE evaluation, the more users' personalized data are revealed. In addition, the multi-tier IoT architectures make security provision difficult. Especially, for vehicular environments in

which the topology of the computing network frequent changes due to mobility, security provision is harder to be achieved compared to other networks.

Additionally, quality assessment in IoT requires further research study. Subjective tests are considered the core part of QoE evaluation for multimedia environments. However, existing subjective approaches that are used to measure QoS, may not be suitable for IoT environments, since (1) it is not feasible to carry out subjective tests for every existing or new IoT application due to their big diversity; (2) they require user feedback after every specific interval, resulting in high network delay and relatively low application response time [23]. In particular, for real time monitoring IoT applications, this can lead to malfunction or in other cases can be dangerous even for humans' safety; (3) they cannot easy determine the cause-roots of the performance, e.g., as stated in [48], subjective results for autonomous vehicles cannot be very helpful for policymakers to define the cause of a car accident; and (4) subjective assessment requires human participation and is usually performed in a (rather isolated) lab environment. Even if we build objective models from subjective tests, their validity will be limited only to the application scenarios for which they are tested [44]. Thus, further study of QoE assessment in IoT is required.

Finally, the conducted research showed that although mulsemedia content provides a new content experience that goes beyond traditional media, QoE evaluation for such types of content is an under-researched area. More QoE IFs should be determined in order to reflect the human-to-machine interaction and, thus, create accurate QoE models. However, the complexity of this task is further increased due to the fact that the majority of the existing olfactory information based systems and methods is only available in specialized laboratories [49]. In addition, there is guidelines on how to create a multisensory content [50], as well as there are not many mulsemedia datasets available.

Table 7 overviews the challenges concerning quality in IoT.

**Table 7.** Quality challenges in IoT.

| Challenge | Existing Solution | Drawback |
|---|---|---|
| Define end-to-end quality in IoT | Use existing QoE definition Define new terms according to the context of the application | It cannot be applied to autonomic IoT Impractical due to the diversity of applications |
| Identify the appropriate IoT IFs | Based on the application different IFs are considered | Impractical due to the diversity of applications |
| Collect IFs Measurements | At different layers from different nodes of the IoT architecture | Increase communication delays There is no unified and inter-operable standard |
| Security and privacy | Based on the application domain different approaches are proposed | Impractical due to the diversity of applications and the increasing appearance of new threats |
| Assess quality | Subjective methods Objective methods | Subjective methods suffer from user bias Objective methods expensive |

## 7. Conclusions

Quality is an important factor in an IoT environment. Quality provisioning in such environments is not only limited to life-threatening situations, but also needs to consider the risk of causing significant business losses and environmental damage [48]. QoE is the most popular metric that has been used to evaluate quality. However, due to fact that initially QoE was introduced to assess the end-user satisfaction, the concepts of traditional QoE should be extended in order to include contextual factors that are important in the IoT domain. In addition, more quality metrics are required in order to evaluate quality in IoT. To this extend, this paper has surveyed the actual necessity of evaluating the quality in IoT. We identified the quality metrics that impact the quality in an IoT environment. However, even

the collection of the quality metrics measurements is not an easy process. For one point, data should be collected from multiple IoT nodes locating at different tiers depending on the IoT architecture that is used, while the storage and transfer of these obtained large-scale data is a very challenging task. Even existing assessing methods should be re-examined in the context of IoT. Especially for mulsemedia applications, traditional QoE assessment methods are not adequate. Thus, research is needed in order to deal with these challenges.

**Author Contributions:** Contributions: Writing—original draft: A.S.; resources: A.S. and P.C. Writing—review & editing: P.C. All authors have read and agreed to the published version of the manuscript.

**Funding:** This research received no external funding.

**Institutional Review Board Statement:** Not applicable.

**Informed Consent Statement:** Not applicable.

**Data Availability Statement:** Not applicable.

**Conflicts of Interest:** The author declares no conflict of interest.

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
