# Peer review of "Defining and Assessing Quality in IoT Environments: A Survey"

_2624-831X, doi:10.3390/iot3040026_

Round 1

Reviewer 1 Report

The paper is interesting and timely.

Almost all figures were copied directly from the sources. They should be redrawn at least.

Chapter 5 Discussion is too brief.

Some references are not correctly inserted in the text.  

There are some typos and editing errors in the article.

Reviewer 2 Report

The authors claim that the existing QoE methods need to be extended to cover QoE within the IoT domain. The authors list extensive studies from the literature.

However, the article lacks novel findings. Computing power, storage capacity, battery life, are all relevant for non IoT devices, such as smartphones, as well. Figure 4 presents a QoE taxonomy for IoT however it is hard to see what is specific and novel, and different from existing work. For example, there are several existing survey studies such as "Kaneez Fizza, et al. QoE in IoT: a vision, survey and future directions. https://link.springer.com/article/10.1007/s43926-021-00006-7" that the authors also cited. However, what is new in this work than the existing?

"Existing QoE evaluation methods for multimedia applications are not suitable for the IoT paradigm." Authors repeat several times in the text the argument that existing methods are not suitable, however there is no new evidence presented in the article that they are not (except the literature). For example, this can be shown with a case study accompanied by results. 

Another reference that can be suggested to the authors is: https://www.ericsson.com/en/reports-and-papers/ericsson-technology-review/articles/machine-qoe-in-the-internet-of-things. In this article, a machine QoE framework is presented that consists of various aspects that are evaluated in categories such as automation, reliability, planning, operation within a case study.

I would like the authors to clarify what the exact contribution of this article is. Is the contribution of the paper summarizing all the challenges, e.g., as in Table 3?

It would help if the authors list the main contribution of this article, where some suggestion would be to summarize all literature work in one Table, make a comparison amongst them stating what is missing in each that this paper is covering. 

Comments on the grammar:

line 4: remove "etc, "

line 20: "catering the quality expectation of the end users in multimedia services is the most important parameter" can be changed to "catering the quality expectation of the end users in multimedia services is important"

line 19: what does the sentence start with "However"? the importance of QoE is mentioned which is a continuation of the previous sentence. 

line 58: "Statista (www.statista.com)", a hyper link would do or reference would do better.

line 79: "i) Applications are executed closer to end-user and IoT devices" why is this an advantage?

line 80:  Latency, response time, and cloud workload for real-time applications are improved" -->  "Latency, response time, and cloud workload for real-time applications are reduced"

line 101: "Eggger" --> "Egger"

Line 135, 140: remove etc.

Line 142: dimensioms --> dimensions

Line 177: typo "besides the user's oe..."

In addition, Figures 1 and 3, 4 are blurry and needs to be fixed with a higher resolution image. Figure 4 is out of margin as well; hence should be aligned properly. 

Overall, there are too many paragraphs in the text. E.g., for every reference, authors use a separate paragraph, even though some paragraphs consist of only one sentence. 

Avoid using "etc." throughout the text and figures (e.g., Figure 3).

Reviewer 3 Report

This manuscript proposes a review of the Quality-of-Experience mechanisms governing the IoT world. Being a survey paper, it does not present any research or technological advance. This notwithstanding, the survey seems to be interesting since it focuses on a very specific part of the field of QoE for IoT. I have just a minor concern regarding the discussion Section (5) which seems to be too short. A survey paper, in fact, should present an extensive discussion of results, considerations, and critical observations concerning the topic. 

Author Response

In the revised version of the manuscript, following the reviewer's comment, the discussion section has been extended.

Round 2

Reviewer 1 Report

The authors have resolved the reviewer's comments. Therefore, the paper should be published.

Author Response

Comment 1.

The authors have resolved the reviewer's comments. Therefore, the paper should be published.

Answer. We would like to thank again the reviewer for his/her valuable comments.

Reviewer 2 Report

Thanks to the authors for putting more time on improving the paper. Although the authors are studying a very important and timely topic, unfortunately, I still have hard times understanding the contribution of this article. An expectation from a survey paper is also in-depth and structured discussion and comparison of existing work systematically along with the  listing of existing work. I would expect to see a detailed and structured summary comparison of all prior art (preferably in one comprehensive Table), and then either clearly pinpoint open challenges that have never been discussed, or to provide a solution to address a set of those challenges. The presented and discussed challenges in the article are known and not non-obvious. One way forward is that the authors focus on only a few aspects of all presented challenges and provide a method to address them that includes the experiment design, user study, and evaluation strategy. For example the authors mention about the potential benefit of using machine learning in the Discussion section, and indeed that could be one good direction and to present how ML can  address the heterogeneity problem in IoT QoE. 

Other comments:

The authors emphasize ML for the first time in the discussion, which could be further studied in the scope of this paper much earlier. 

Figure 1 is partially taken from the following paper (with citation), but x-axis is missing which yields confusions. https://www.ncbi.nlm.nih.gov/pmc/articles/PMC5191038/

"We present and discuss the different approaches that classify the QoE IFs, focusing more to the data and information parameters that are necessary for the creation of a successful QoE model."  It is hard to understand what this sentence means.  

Figure 6 picture quality is poor. Any possibility to improve? In addition what do the replicated "Satisfaction" boxes stand for?

Typo in line 185: "to accurate predict the QoE of"

Table 4: what does testing column stand for? There is no explanation. 

Typo in line 209: "sensitivity have on 360 mulsemedia

Author Response

Comment 1.

Thanks to the authors for putting more time on improving the paper. Although the authors are studying a very important and timely topic, unfortunately, I still have hard times understanding the contribution of this article. An expectation from a survey paper is also in-depth and structured discussion and comparison of existing work systematically along with the listing of existing work. I would expect to see a detailed and structured summary comparison of all prior art (preferably in one comprehensive Table), and then either clearly pinpoint open challenges that have never been discussed, or to provide a solution to address a set of those challenges.

Answer 1.

Following the reviewer ‘s comment we have revised the paper accordingly. More specifically, we have added 3 tables (Tables 1, 5 and 7) that either compare existing solutions (tables 1 and 7) or highlight and list existing works (table 5).

Comment 2.

The presented and discussed challenges in the article are known and not non-obvious. One way forward is that the authors focus on only a few aspects of all presented challenges and provide a method to address them that includes the experiment design, user study, and evaluation strategy. For example the authors mention about the potential benefit of using machine learning in the Discussion section, and indeed that could be one good direction and to present how ML can address the heterogeneity problem in IoT QoE.

Answer 2. Following the reviewer ‘s comment the discussion section has been revised accordingly. We added a solution on how ML can be used to identify quality IFs in IoT. Moreover in table 7 we highlighted the challenges, existing solutions, as well as, their shortcomings.  

Round 3

Reviewer 2 Report

Hi, 

Thanks to the authors for addressing the comments, and for improving the document quality significantly. The latest version reads much better. There are still a few minor fixes required; mainly in aesthetic improvements in the figures. Figure 5 is too large, and can be improved. Figure 6 is hardly readable, but a better choice of colors in the figure should fix it without so much effort. In Figure 1, some content is on the right, while others on the left although there is no x-axis in the figure. Hence the figure itself can be simplified. In addition, in Figure 1, there are some rectangles with inconsistent color coding (Environmental monitoring vs Smart parking). 

Conclusion consists of too many short paragraphs(some with one sentences). It can be better formatted to ease readability and aesthetics.   
